# A Wideband and High-Power RF Switching Design

**DOI:** 10.3390/s25103209

**Published:** 2025-05-20

**Authors:** Xindong Huang, Chengying Chen, Shaokang Zhou

**Affiliations:** School of Opto-Electronic and Communication Engineering, Xiamen University of Technology, Xiamen 361024, China; huangxindong@xmut.edu.cn

**Keywords:** RF switches, wide operating bandwidth, isolation

## Abstract

This paper presents an RF switch chip with a wide operating bandwidth from 6 to 18 GHz, designed for RF front-end applications in mobile communications. A series-parallel topology combined with a stacked transistor structure was employed to improve power handling while maintaining low insertion loss and high isolation. To further optimize isolation and return loss, LC resonant circuits were introduced by utilizing off-state transistors as capacitive elements. Compared to existing designs, the proposed switch achieved an improved trade-off between bandwidth, power capacity, and port performance. Measurement results showed insertion loss below 1.917 dB, isolation above 38.839 dB, return loss better than 13.075 dB, and 1 dB input compression point above 32 dBm at 12 GHz, confirming the effectiveness and novelty of the broadband design.

## 1. Introduction

Radiofrequency (RF) front-end chips represent a crucial component of modern wireless communication systems. The demand for high performance has propelled significant advancements in semiconductor technology.

The early transceiver (T/R) systems were complex and costly to manufacture. With the advent of semiconductor technology, microwave-integrated circuits (MICs) have gradually evolved into monolithic microwave-integrated circuits (MMICs). In the present era, several multifunctional MMIC chips can be integrated to form a comprehensive RF transceiver front-end system, as evidenced by the literature [1,2,3]. RF front-end chips integrate a variety of structures, including RF switches, low-noise amplifiers, and power amplifiers.

In 2016, scholar Nguyen designed a high-power RF switch operating at 22–26 GHz based on a 0.15 μm GaAs process. The switch utilizes a quarter-wavelength line in parallel topology. The switch employs a stacked transistor configuration with capacitors in series, thereby enhancing voltage handling capabilities. Ultimately, this configuration enables the switch to achieve an input 1 dB compression point of 36 dBm in the K-band. However, due to the stacked structure and the introduction of series capacitance, the switch experiences an additional signal loss in the on state, negatively affecting the insertion loss performance. The final test results indicate that the switch exhibits an insertion loss of 2.5 dB within the operational band [4,5]. This drawback is also evident in the RF switches designed by Peng-I Mei and Yi-Fan Tsao, who similarly employed the stacked structure to augment the power handling capacity of the switches, thereby enhancing the overall power handling capability of their switches to 26 dBm and 36 dBm, respectively. However, the insertion loss of these switches is higher. The switch designed by the former author exhibits an insertion loss of 3 dB within the 15 GHz to 35 GHz range [6]. The insertion loss of the switch designed by the latter author reaches 3.2 dB within the 12 GHz to 40 GHz band [7].

In 2022, Scholar Ze Fan devised a low insertion loss, high-power single-pole, double-throw series RF switch based on a 0.15 μm GaAs PHEMT process. Utilizing capacitors and microstrip lines, Ze Fan constructed a π-equivalent circuit, which effectively shortened the transmission distance of RF signals and reduced the insertion loss. A stacked configuration was also employed to enhance the switch’s ability to withstand high-power levels. The final test results demonstrated that the power handling capability of the switch reaches 32.44 dBm in K-band with an insertion loss of less than 1.14 dB. However, the minimum isolation of the switch is only 15 dB, which results in high-power signals at the shutdown port when processing high-power signals. Consequently, the switch is unsuitable for RF transceiver front-end applications [8].

In recent years, some RF switch chips designed based on the second-generation semiconductor process have also been reported in China [9,10,11,12,13]. 2021 In 2021, Establishment Xin designed a multifunctional MMIC RF front-end chip based on the 0.15 μm GaAs process for 6–18 GHz, which contains a power amplifier, a low-noise amplifier, and a transceiver switch, and the transceiver SPDT adopts a series-parallel type of asymmetric structure, which can withstand a 30 dBm power signal [14]. Kefan Chen, Fengjun Chen, et al. designed a low insertion loss, high isolation single-pole, double-throw switch based on the 0.25 μm GaAs PHEMT process in 2019. The switch exhibits an insertion loss of less than 1.4 dB and isolation greater than 35 dB in the 0-20 GHz band. However, its input 1 dB compression point is 27 dBm, which narrows the application range of the chip [15]. In 2022, Wenjie Lv designed a high isolation RF switch based on 0.5 μm GaAs process, innovatively loaded with RC parallel structure at the source level of PHEMT, which effectively improved the isolation degree; the simulation results showed that the isolation degree of the switch is better than 35 dB, the insertion loss is less than 1 dB, and the power handling capability at 12 GHz is 21 dBm, and a switching time of nanoseconds [16].

From the current development of RF switches, it can be found that the RF switch performance affects each other. When too much emphasis is placed on a certain performance, it will reduce the other performance of the RF switch, which requires trade-offs between the different performances of the RF switch in the design process. This thesis focuses on broadband RF transceiver switches and studies improving their power handling capability according to the system’s index requirements.

The chip in this paper is an RF switch chip based on 0.25 µm GaAs PHEMT process, operating at 6–18 GHz, with a chip area of 1200 µm × 800 µm. During the design process, this paper strictly followed the design flow of RF switches and, firstly, according to the given design indexes, put forward the two design difficulties of the switch. For the difficulties in the operating bandwidth, we designed the core of the one-string-two-parallel structure. The topology of a one-series-two-parallel structure was designed to address the difficulty of wide operating bandwidth, and the stacked transistor structure was proposed to address the difficulty of high-power processing capability. Subsequently, the preliminary schematic diagram of the chip was drawn according to the designed structure, and the preliminary schematic’s isolation and return loss performance was optimized by using the functional series transistor and the functional parallel transistor. Finally, the schematic and layout of the chip were designed, and simulation and testing were completed.

Compared to existing RF switch designs, most prior work either focuses on narrowband performance or lacks sufficient power handling over wide frequency ranges. In contrast, this work provides a compact solution that achieves a superior balance of insertion loss, isolation, return loss, and power tolerance over an ultra-wide 6–18 GHz band. The combination of a series-parallel switching topology and resonance-based performance optimization distinguishes this design from conventional approaches.

The key contributions of this work are as follows:

A wideband RF switch chip with a novel series-parallel topology was proposed that supports stable operation from 6 to 18 GHz.

A stacked transistor structure was implemented to significantly improve the power handling capability.

LC resonance techniques were employed to improve both isolation and off-state return loss.

The fabricated chip demonstrated high performance with low insertion loss, excellent isolation and high linearity, validated by measurements.

## 2. Materials and Methods

The reflection coefficient, denoted by γ, is to be calculated by the method outlined in Equation (2). The return loss performance of the transceiver switch is typically expressed in terms of the lowest return loss of each port within its operational bandwidth.

The common topologies for RF switches are series, parallel, and a combination of the two series-parallel RF switches. Series-type switches have very good isolation and insertion loss performance at low frequencies. Due to the on-resistance and cutoff capacitance, the frequency change is very sensitive to the switch impedance: when the frequency increases, the on-resistance rises, and the cutoff impedance falls, so the series-type RF switch is unsuitable for high-frequency applications.

Parallel-type structure RF switches are only suitable for narrowband applications, and for low-frequency applications, the quarter-wavelength transmission line is very long; for example, the length of the quarter-wavelength transmission line for 6 GHz under this process reaches 1.8 mm, and the quarter-wavelength for 18 GHz also reaches 0.6 mm, which consumes a large amount of chip area and raises the cost of fabricating the chip, so the parallel-type RF switches are generally used for high-frequency scenarios with shorter wavelengths.

The structure of the series-parallel-type switch is shown in Figure 1, which has a wide operating band. The two branches of the series-parallel RF switch in Figure 1 are completely symmetrical, and the DC power sources VDC1 and VDC2 control the opening and closing of the signal path. Microstrip lines TL1, TL2, and TL3 are used to improve the matching of the individual ports. Each transistor is resisted in series at the gate to minimize RF signal leakage and reduce the insertion loss of the switch.

When dealing with high-power signals, the effect of the signal on the pathway must be considered, and excessive reflections may cause irreversible damage to the circuit. In the structure of RF switches, they are categorized into reflective and absorptive switches according to the matching of the output port in the closed state. Reflective switches have open or shorted output ports when closed, and all RF signals are reflected, which may result in high voltages that can damage connected devices. In contrast, an absorptive RF switch absorbs the input power signal to avoid damage to the preamp circuitry from high-power reflected signals and is more suitable for high-power applications. Figure 2 illustrates an absorptive switch where PORT1 is on and PORT2 is off when V1 is high, and V2 is low. If the transistor is an ideal switch, the input impedance of PORT2 is close to 50 Ω, which ensures good circuit matching and avoids reflection of high-power signals, thus protecting the device. Therefore, this chip is designed as an absorptive RF switch to meet the performance requirements of high-power signal processing.

After selecting the topology as a series-parallel-type topology, the number of series and parallel tubes must be considered as it directly affects the RF switch’s insertion loss and isolation performance. Figure 3 shows the insertion loss and isolation performance at 1–20 GHz for three structures of RF switches, a series-parallel, a series-two-parallel, and a series-triple-parallel, where the dimensions of the transistors used are 4 × 135 μm PHEMTs and the bias voltages are 0 V and −5 V. The insertion loss and isolation performance of the three structures of RF switches are based on the number of tubes in series and parallel.

Figure 3 shows that as the number of parallel tubes increases, the isolation performance of the RF switch improves over the whole frequency band. However, the equivalent impedance of the cutoff transistor decreases at high frequencies, and the insertion loss increases significantly as the number of parallel tubes increases. The insertion loss of the RF switch with one string and two parallel structures in Figure 3a reaches 3.5 dB at 18 GHz, which is more than the 2 dB required by the design specifications, so it is necessary to optimize the size of the transistors to reduce the insertion loss of the switch during design.

The output power expression of the switch is as follows:(1)$$Pout=vo^{2}/Z0$$(2)$$Pout=io^{2}\times Z0$$

$P_{out}$ is the output power of the switch, $i_{o}$ and $v_{o}$ are the output voltages, and $Z_{0}$ is the port load, which is typically 50 Ω. According to the specification requirement, the RF switch should have more than 1 W power handling capability. From Equations (1) and (2), it can be seen that $i_{o}$ and $v_{o}$ should be greater than 0.14 A and 7.07 V, respectively, where voltage and current are RMS values. For the SPDT switch shown in Figure 1, M2 and M3 operate in the linear region when PORT2 is activated. When the signal input current increases to the saturation current of M2, the operating state of M2 changes from the linear region to the saturation region. At this time, the source-drain voltage increases, and the signal output power is compressed according to Equation (1). If the signal voltage swing is too large and M3 changes from the off state to the on state, M3 will consume the RF current, reducing the output current and further compressing the output power according to Equation (2). Therefore, the power handling capability of a series-parallel-type switch is jointly determined by the operating states of the series and parallel transistors. To increase the power handling capability of the switch, the current handling capability of the series transistor and the voltage handling capability of the shunt transistor must be increased.

Increasing the size of the series transistor can improve its saturation current value, but unquestioningly increasing the size can lead to serious parasitic effects. Figure 4 shows the values of the transistor parasitic parameters in the on and off states for different sizes; the larger the size, the more pronounced the parasitic effect, which causes difficulties in the broadband design of the switch. Therefore, the transistor size should be reduced as much as possible while maintaining the power handling capability to reduce the influence of parasitic effects.

To ascertain the minimum size of the series transistors, the source-drain voltages and currents of the transistors at different sizes are simulated, as illustrated in Figure 5. As illustrated in that figure, when the VDS is minimal, the transistor size reaches 4 × 100 µm to achieve a current handling capability of 0.2 A. Consequently, the transistor size must be a minimum of 400 µm to enable the RF switch to have a power handling capability of 1 W or greater.

The stacked structure of the parallel transistors (shown in Figure 6a) improves voltage handling in the off state, with breakdown voltages up to twice that of a single transistor. As the higher gate resistance isolates the two gates, the input voltage is evenly distributed across capacitors *Cgd1*, *Cgs1*, *Cgd2*, and *Cgs2*, reducing the gate-source voltage of the stacked transistor to half that of a single-layer transistor. As a result, the power handling capability of a stacked RF switch can be increased by 6 dB.

The stacked-FET structure is shown in Figure 6a, Since $R_{1}$ and $R_{2}$ isolate the transient gate, the $V_{rf}$ will be equally distributed by the $C_{gd1}$, $C_{gd2}$, and it shrinks the gate source voltage of $M_{1}$ and $M_{2}$ to half that of the single-layer transistor. Figure 6b shows the instantaneous values of the voltages $V_{g1}$ and $V_{g2}$ with $V_{rf}$. However, blindly stacking the number of transistors deteriorates the insertion loss and isolation performance of the switch (Figure 7).

Figure 6b shows the instantaneous values of $V_{g1}$ and $V_{g2}$ as a function of Vrf. According to the process library file provided by the foundry, it is known that the on-state voltage of the switching tube is −1 V, so the equation for the signal voltage processing capability using a two-layer stacked transistor, as shown in Figure 6a is expressed as follows:(3)$${V_{DC}+V_{g1}-V_{s1}<V}_{TH}$$
where $V_{DC}$ is −5 V, $V_{g1}$ and $V_{s1}$ are 3/4 and 1/2 of the RF signal, respectively, and the inequality reduces to(4)$$V_{DC}+\frac{Vrf}{4}<V_{TH}$$

Calculations show that Vrf can reach a maximum of 16 V. At the same time, the voltage handling capability of a single-layer transistor is 8 V. Since the power handling capability calculation is based on the RMS value of the voltage, a single-layer transistor will not allow the RF switch to achieve 1 W of power handling capability. To handle 1 W of power, the RF switch must be able to handle a 10 V sinusoidal peak voltage, so at least a two-layer stacked transistor design is required. However, increasing the number of transistor layers will result in higher insertion loss. Figure 3a shows that the insertion loss increases significantly as the number of stacks increases. This design uses two layers of stacked transistors to balance power handling, insertion loss, and system cost.

In a series-two-parallel structure, both series and parallel transistors can process voltage and current signals as the switches are turned on and off, making the design of transistors at key nodes critical. For the chip to achieve a power handling capability of more than 1 W, the current must be larger than 400 µm, while the transistors handling the voltage require a two-layer stacked structure. Figure 8 shows a preliminary schematic of chip A, where M1 and M2, M7 and M8 are series transistors, and the other transistors form a two-layer stacked parallel transistor. The working principle will be analyzed next.

When VGS1 is −5 V and VGS2 is 0 V, transistors M1, M2, and M9–M12 conduct, and the rest of the transistors turn off, the PORT2 port turns on, and the PORT1 port turns off. If PORT2 flows into a high-power signal, M1 and M2 will flow through a high-power current; the signal generates a high-power voltage at the PORT1 port, and the series tubes connected to the off state here need to isolate the high-power voltage. Both M7 and M9 need to deal with the large voltage swing at the time of switching off, so M7 and M9 use a two-layer stacked transistor. Similarly, when VGS1 is 0 V and VGS2 is −5 V, M1 and M2 must also handle large voltage swings and use a stacked transistor structure.

According to the simulation results in Figure 3a, the insertion loss of the RF switch with a one-string-two-parallel structure rises from 0.5 dB to 2.2 dB, and the isolation decreases from 40 dB to 29 dB in the 6–18 GHz band, which fails to meet the design requirements. In addition, the switch shutdown port has no absorptive design, which may damage other circuits. Therefore, the schematic must be optimized for isolation and absorptive structure design.

To optimize the isolation performance of the switch, a functional series transistor, as shown in Figure 9a, has been integrated into the circuit schematic. Its equivalent circuit in the cutoff state is depicted in Figure 9b. As shown, the equivalent model forms an LC parallel resonant circuit that presents maximum impedance at its resonant frequency (defined as $1/\sqrt{\mathrm{LC}}$). The following analysis discusses the various operating states of the transistor and the role of the overall circuit configuration.

When the transistor is turned on, it exhibits very low conduction impedance, allowing the RF signal within the operational frequency band to pass primarily through the transistor branch. This configuration slightly reduces the insertion loss of the RF switch.

Conversely, when the transistor is in the cutoff state, a shunt resonant circuit is formed by its cutoff capacitance and parallel inductance. This parallel resonance results in a high impedance at the resonant frequency. By tuning this resonance to the mid-frequency range of 6–18 GHz, the switch can achieve enhanced isolation across the operating band.

Figure 10 shows the simulation of the change in switch performance with the addition of functional series tubes. A comparison of the insertion loss and isolation of the switch before and after the introduction of the functional series tube is shown in Figure 10. Figure 10a shows that the insertion loss of the switch increased by about 0.2 dB at each frequency point due to the increase in circuit losses caused by the increase in the number of devices conducting the pass-through path. Despite the increase in insertion loss, the functional tandem tube significantly improves the isolation of the switch, and the results in Figure 10b show a clear upward convex trend in isolation, indicating a significant improvement in isolation performance.

The return loss performance of RF switches is usually better for conductive ports because the ideal conductive pass-through has no signal loss between the input and output ports, and the ports are perfectly matched. However, for shutdown ports, matching the signal ports determines whether the switch is reflective or absorptive. With a reflective switch in the off state, the output port is either open or short-circuited, and the RF signal is fully reflected. On the other hand, absorptive RF switches absorb the incoming power signals from the preamplifier circuits when switched off, thereby avoiding the generation of high-power reflected signals that are harmful to the preamplifier circuits. Therefore, high-power RF switches must be designed with appropriate matching circuitry.

Figure 11a shows the absorbing structure introduced. With the port closed, the equivalent circuit of the transistor in this structure is shown in Figure 11b, which behaves as an LC series resonant circuit, where R1 is used to reduce the Q of the circuit.

Figure 12 shows the simulation results of the change in return loss at the turn-off port by adding the functional shunt tube. The figure shows that the turn-off port has a significant dip at the resonance point, indicating that the introduction of the functional transistor dramatically improves the matching of this port in this frequency band.

After adding the isolation optimization and absorbing structure, the schematic of this chip is shown in Figure 13. We used color coding techniques to clearly highlight the signal paths and improve the visual clarity of the crossed routes. The core topology in the schematic consists of M1–M6 and M10–M15 symmetrically connected in a series and two parallel structures. The functional transistors are M9 and M18 in series and M7, M8, M16, and M17 in parallel, where transistors M9 and M18 are connected in parallel with inductors L1 and L2 to improve the impedance of the turn-off path, i.e., M7, M8, M17, and M18 are connected in series with capacitance and a resistor to optimize the output impedance of the shutdown path in the range 6–18 GHz, which improves the matching return loss performance of the port. TL1–TL3 on the ports are the port-matching microstrip lines, and TL4–TL7 are the longer microstrip lines connecting the devices.

The dimensions and values of the main devices of the final chip are shown in Table 1.

## 3. Results

### 3.1. Graphic Design

The configuration is illustrated in Figure 14, and the devices depicted in the layout match precisely those shown in the schematic. The layout dimensions are 1200 µm × 800 µm, encompassing an area of 0.96 mm^2^. This meets the chip area requirement as outlined in the design specification. The PORT2 and PORT3 port snakes in the layout serve the purpose of increasing port matching.

### 3.2. Pattern Simulation

The joint simulation process proceeds as follows: after completing the electromagnetic (EM) simulation, a new model is instantiated in the layout. A new cell is then created and assigned as the EM simulation model. The original transistor schematics are removed, and the two schematic domains are connected. The resulting schematic and layout are shown in Figure 15, where all components—except for the active devices and test control circuitry—are derived from the EM simulation model.

After the joint simulation diagram has been created, the joint simulation diagram is simulated, at which time the simulation results are shown in Figure 16.

As shown in Figure 17, the return loss of the switch in the 6–18 GHz band is higher than 13.8 dB, the insertion loss of the switch is less than 1.989 dB, and the isolation is higher than 35.7 dB. Compared with the schematic simulation results in Figure 13, the return loss of the switch decreases by 1.2 dB, the insertion loss increases by 0.6 dB, and the isolation decreases by 1.5 dB. Although the joint simulation results showed a decrease in the small signal parameter performance of the RF switch compared to the schematic results, the layout design specifications met the design requirements.

Figure 17 shows the joint simulation results of the input and output power of the switch at 12 GHz. As can be seen from Figure 17, the switch undergoes power compression when the input power reaches 31 dBm, at which point the insertion loss of the switch is 2.253 dB and does not rise to 2.665 dB (the minimum value of the I.L. simulation is 1.665 dB). Therefore, the switch’s 1 dB input compression point is more excellent than 31 dBm, which meets the design specification.

### 3.3. On-Chip Testing

On-chip testing represents the phase of the testing process whereby the performance of an unpackaged die is evaluated. The probes utilized in this design are GSG probes with a port spacing of 150 µm, capable of testing signals up to 40 GHz. The chips in this design operate within a frequency band below 18 GHz, which aligns with the specifications of the test probe. Consequently, the pad spacing was set to 150 µm during the two-chip layout design. The GSG ports of the chips are staked out using the probe station during the test. Figure 18 depicts the test staking diagram of the chips in this paper, while Figure 19 illustrates the on-chip test environment.

### 3.4. Assembly Testing

The chip’s power processing capability index reached 30 dBm in this paper. However, due to the equipment’s limitations, it was impossible to conduct high-power signal testing directly on the chip. Therefore, this thesis employs power amplifiers and attenuators on the chip for indirect high-power testing. The test block diagram is illustrated in Figure 20. The signal source initially transmits a low-power signal, which is then amplified by the power amplifier on a specified number of occasions to produce a high-power signal. This signal then flows into the RF switch, which the attenuator attenuates on a defined number of occasions, as measured by the power meter. Subsequently, the attenuator attenuates the high-power signal a specified number of times, after which it is measured by the power meter.

Before conducting the assembly test, creating the chip’s assembly circuit board is essential. The designed PCB board is illustrated in Figure 21a. In this study, the chip’s assembly board is a four-layer board constructed using the mixed-pressing method, with a total thickness of the dielectric substrate Rogers RO4003C, eight mils, and the copper cladding is 1 oz. The surface of the exposed copper is made of gold, which has been treated by immersion. To calibrate the impact of the assembly test, it is necessary to subtract the final test results from the pass-through measured loss, which includes the three RF ports and LON, RON, two DC control ports, and the lower part of the pass-through design. This is in addition to the two PCB boards, RFOUT, RFINL, and RFINR.

The chip is bonded using gold wire upon completion of the PCB board manufacturing process, as illustrated in Figure 21b. This bonding method, known as double-wire bonding, can reduce the impact of bonding line parasitic parameters. The assembly test environment is depicted in Figure 21c.

The on-chip test results are shown in Figure 22, from which it can be seen that the insertion loss of chip A is less than 1.917 dB, the isolation is more significant than 38.839 dB, the return loss of the off port is more critical than 13.075 dB, and the return loss of the on port is more incredible than 15.305 dB in the 6–18 GHz frequency band. The maximum value of the operating frequency band is indicated by the triangles in Figure 22.

The ideal bond line added in the simulation has different parasitic parameters than the actual bonding, causing some resonance point shifts.

After deducting the PCB’s straight-through loss, the amplifier’s gain, and the attenuator’s attenuation, the test simulation results are shown in Figure 23, which shows that the input 1 dB compression point is approximately 32 dBm.

A comparison of the test results with the results of the literature published in recent years is shown in Table 2, which shows that this chip’s isolation performance and power handling capability are at a high level in the operating frequency band. The isolation performance is more than 3.8 dB higher than the highest in the listed literature, and the power handling capability is 5 dB higher than the highest in the listed literature, which indicates that the switching chip designed in this paper is more suitable for high-power transceiver systems. Although the designed switching chips all use a stacked series-parallel structure, which increases the number of devices, the chip area is similar to that of the switching chips designed in other literature, and the manufacturing cost is close.

**Table 2 sensors-25-03209-t002:** Comparison of test results with the performance of literature published in recent years.

Parameters	Literature [15]	Literature [10]	Kinds of Literature [16]	Chips in This Article
Process (GaAs)	0.25 μm	0.25 μm	0.5 μm	0.25 μm
Circuit topology	Return loss (dB)	Return loss (dB)	Return loss (dB)	data
Operating band (GHz)	DC-18	DC-15	DC-18	DC-18
Insertion Loss (dB)	1.4	2	1.1	1.917
Degree of separation (dB)	25	30	35	38.839
Return loss (dB)	15.6	15	10	13.076
Pin1 dB (dBm)	27@5GHz	23@5GHz	21@9GHz	32@12GHz
Physical size	1.2 mm × 1.2 mm	1 mm × 1 mm	0.84 mm × 0.82 mm	1.2 mm × 0.8 mm

## 4. Discussion

In this paper, an RF switch chip based on a 0.25 µm GaAs PHEMT process was proposed, which operates in the frequency range of 6–18 GHz with a compact chip area of 1200 µm × 800 µm. Co-simulation results indicated that the chip achieves a return loss better than 13.8 dB, insertion loss less than 1.989 dB, and isolation higher than 35.7 dB, with a power handling capability up to 31 dBm, meeting the design requirements. After fabrication and on-chip testing, measurement results showed a maximum insertion loss of 1.917 dB, minimum isolation of 38.839 dB, return loss better than 13.075 dB, and an input 1 dB compression point of 32 dBm at 12 GHz, validating the performance of the proposed design.

Despite these promising results, several limitations remain. For example, performance may degrade slightly near the edges of the frequency band due to parasitic effects, and the current work does not include a temperature-dependent reliability analysis. Nevertheless, the chip’s wide operating bandwidth, high isolation, and compact form factor make it suitable for practical applications in broadband communication systems, radar, and electronic countermeasures. Future work will focus on integrating control logic, further reducing insertion loss, and improving robustness under varying environmental and packaging conditions.

## Figures and Tables

**Figure 1 sensors-25-03209-f001:**
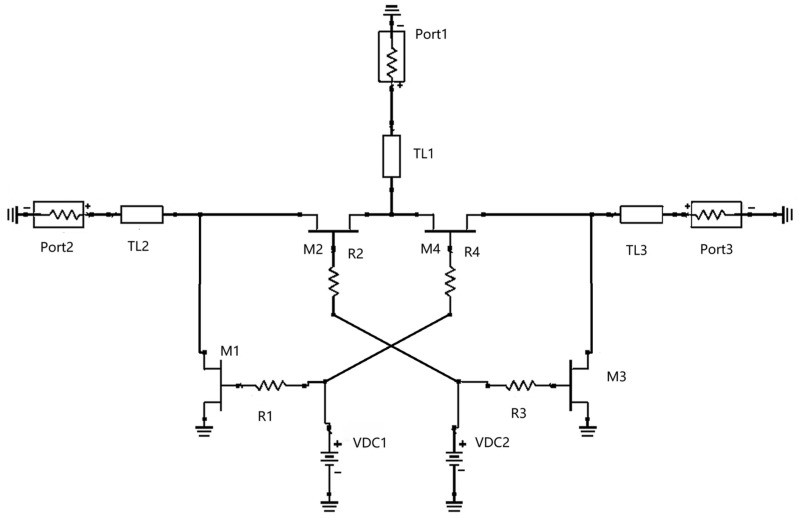
Series-parallel SPDTs.

**Figure 2 sensors-25-03209-f002:**
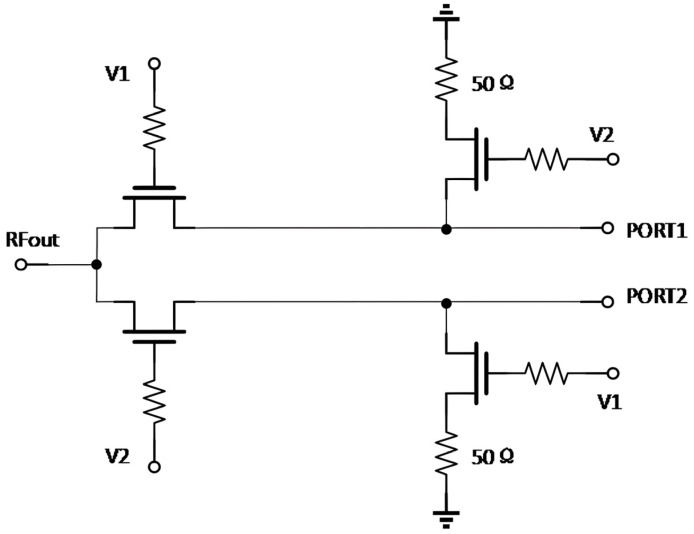
Absorption RF switch.

**Figure 3 sensors-25-03209-f003:**
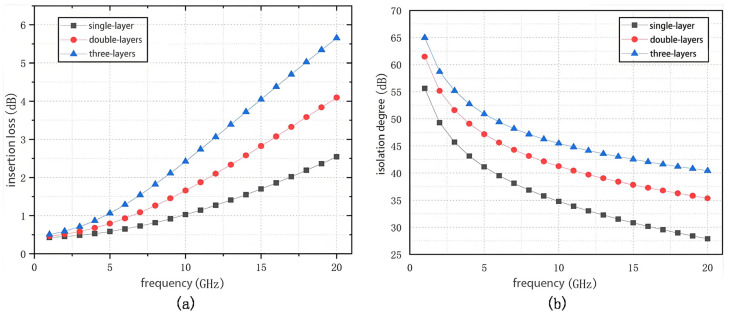
Simulation results of insertion loss versus isolation for multilayer tubes: (**a**) insertion loss; (**b**) isolation.

**Figure 4 sensors-25-03209-f004:**
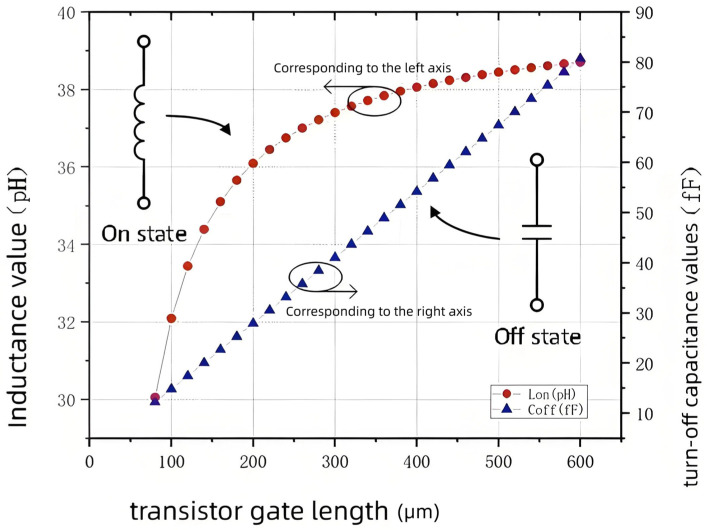
Simulation results of inductance and turn-off capacitance values for transistors of different sizes.

**Figure 5 sensors-25-03209-f005:**
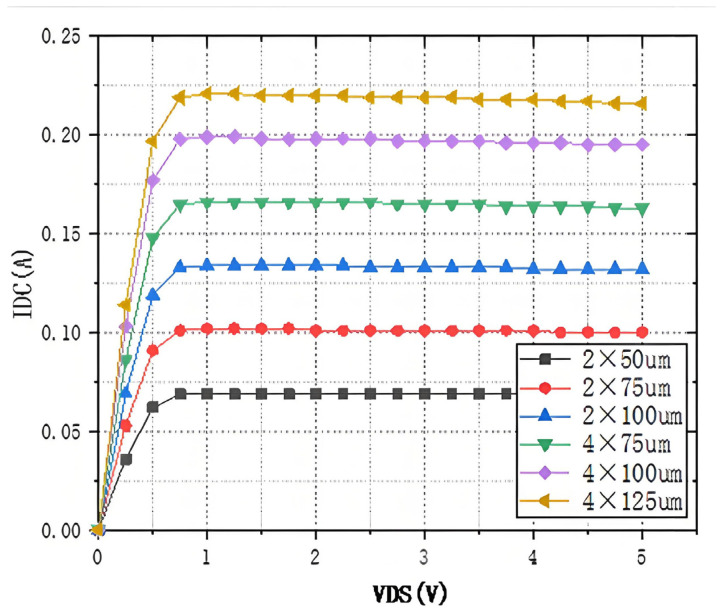
Simulation results of leakage current versus source-drain voltage for different size transistors.

**Figure 6 sensors-25-03209-f006:**
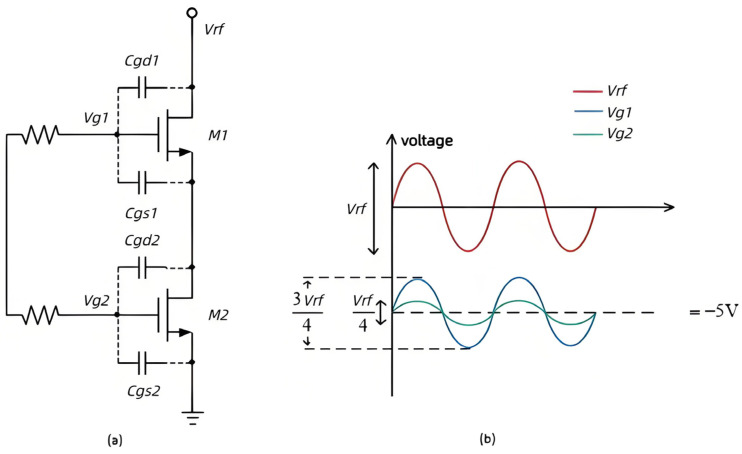
Two-layer stacked transistor structure with primary node voltage: (**a**) structure; (**b**) primary node voltage.

**Figure 7 sensors-25-03209-f007:**
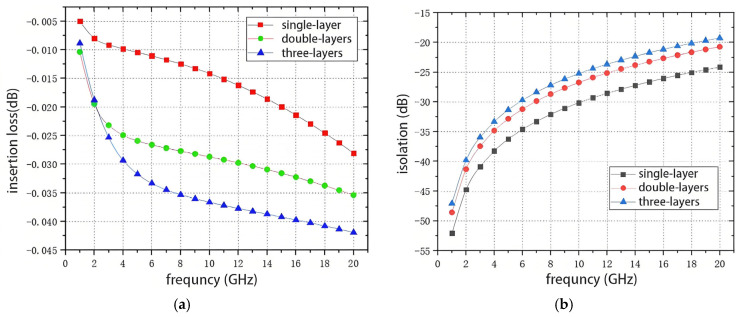
Comparison of various configurations of series-stacked transistors, with consideration of insertion loss (**a**) and isolation (**b**), is presented.

**Figure 8 sensors-25-03209-f008:**
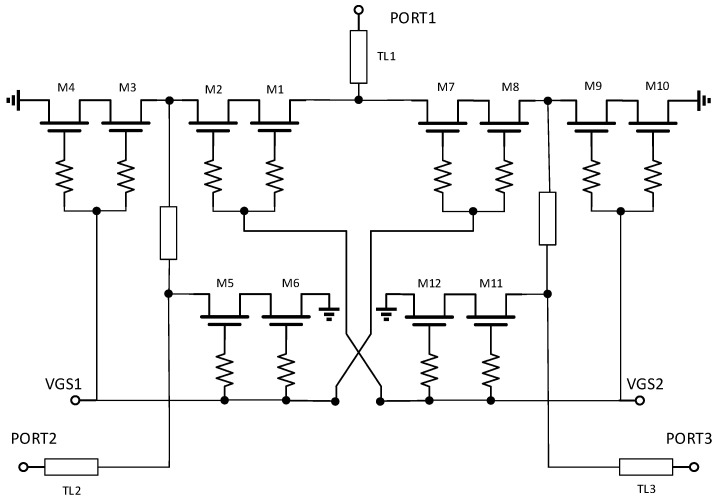
Tentative diagram of the chip.

**Figure 9 sensors-25-03209-f009:**
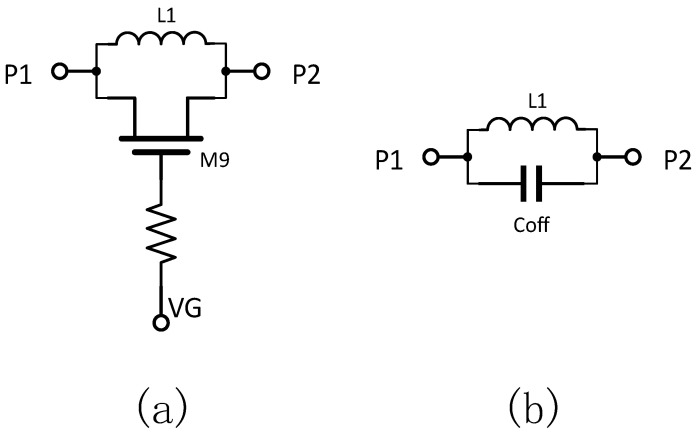
Functional tandem tube structure and its operating equivalent circuit: (**a**) structure; (**b**) operating equivalent circuit.

**Figure 10 sensors-25-03209-f010:**
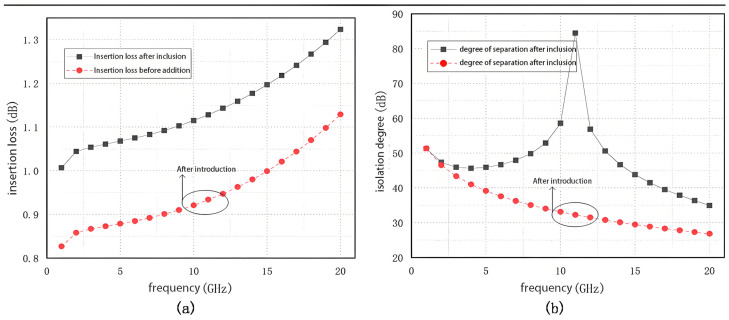
Changes in switching performance with the addition of functional series tubes: (**a**) Insertion loss simulation results; (**b**) Isolation simulation results.

**Figure 11 sensors-25-03209-f011:**
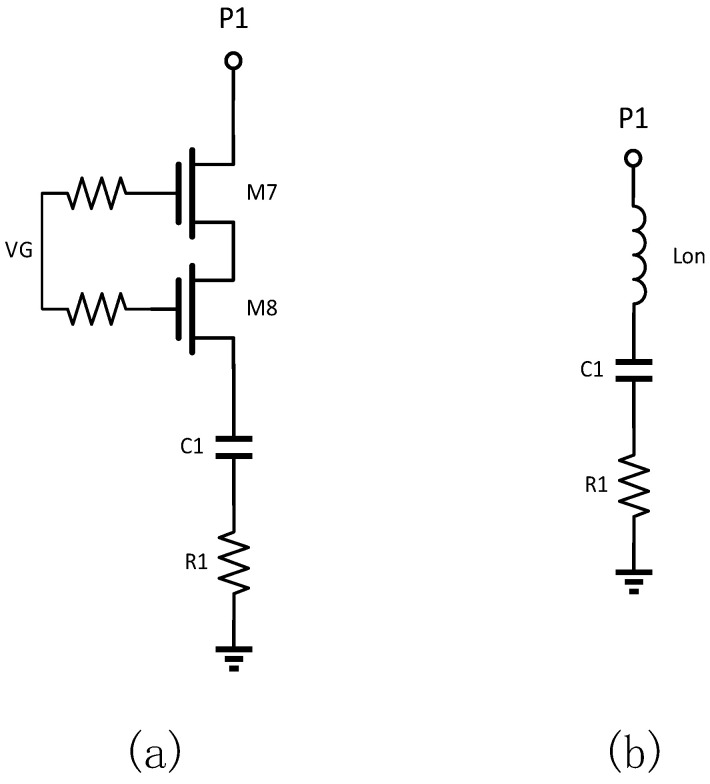
Functional shunt tube structure and its equivalent circuit during operation: (**a**) structure; (**b**) equivalent circuit.

**Figure 12 sensors-25-03209-f012:**
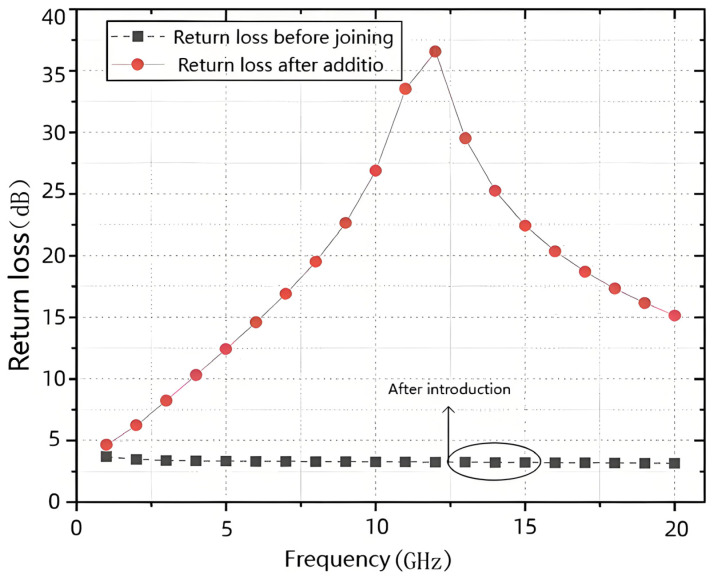
Simulation results of return loss variation at the shutoff port after adding a functional shunt tube.

**Figure 13 sensors-25-03209-f013:**
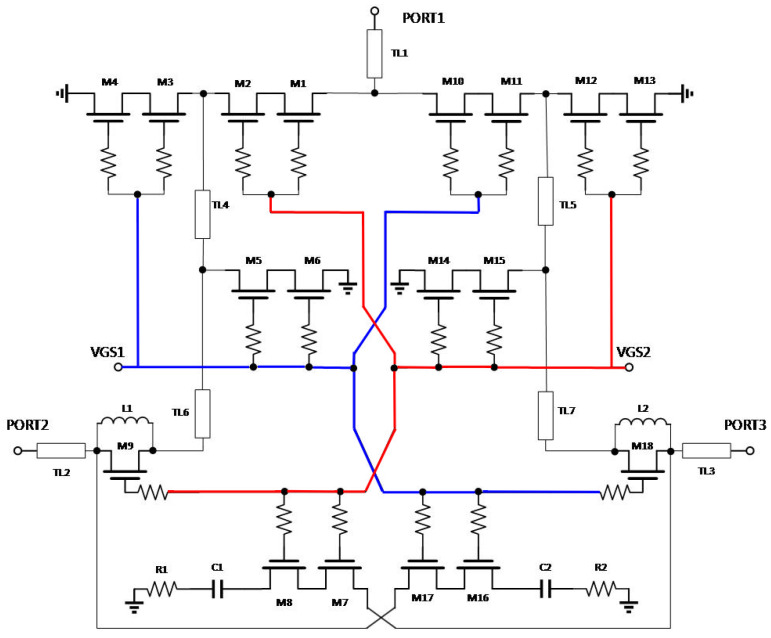
Schematic diagram of the RF switches for the 6–18 GHz range.

**Figure 14 sensors-25-03209-f014:**
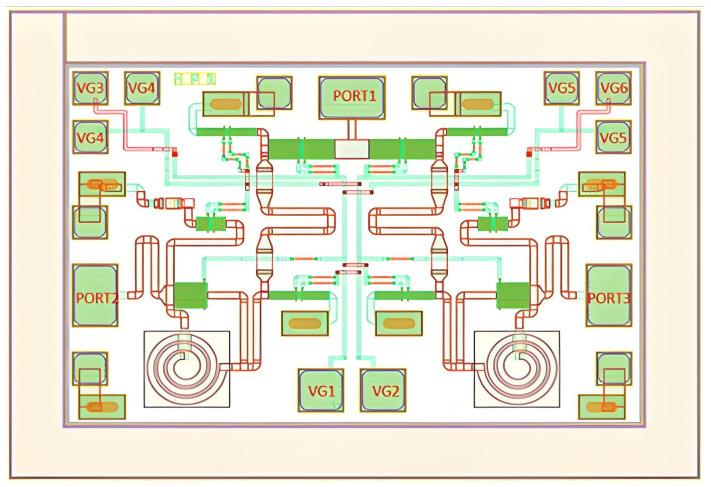
6–18 GHz RF switch layout.

**Figure 15 sensors-25-03209-f015:**
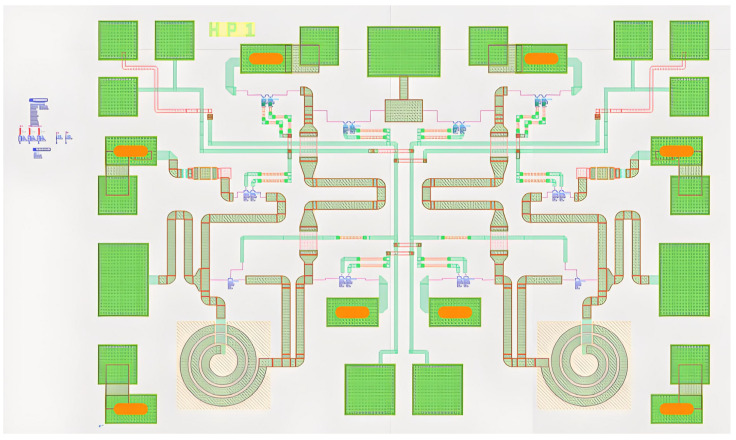
Co-simulation schematic.

**Figure 16 sensors-25-03209-f016:**
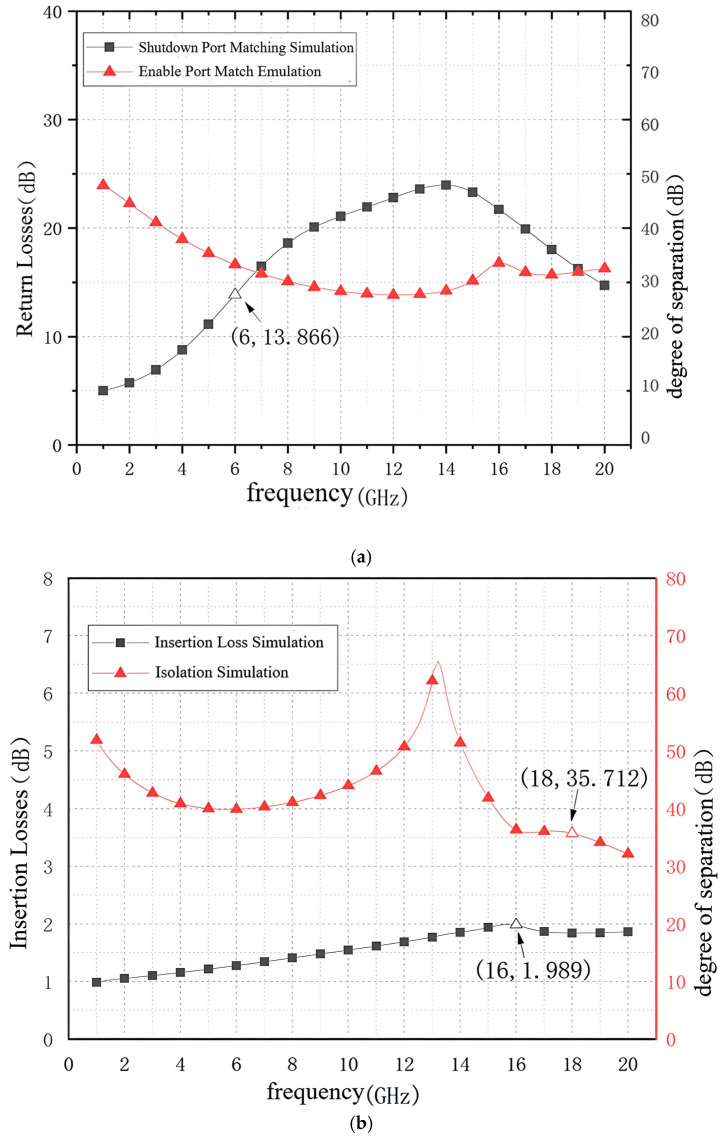
Joint simulation results: (**a**) Return loss; (**b**) Insertion loss vs. insulation.

**Figure 17 sensors-25-03209-f017:**
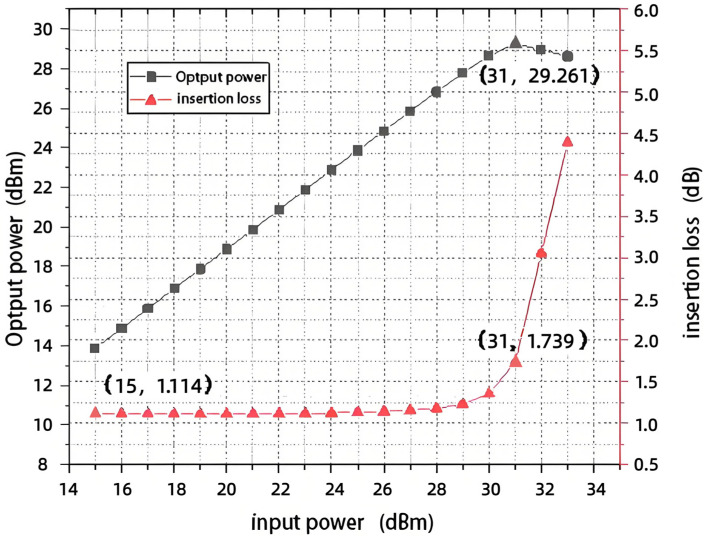
Joint simulation of input and output power.

**Figure 18 sensors-25-03209-f018:**
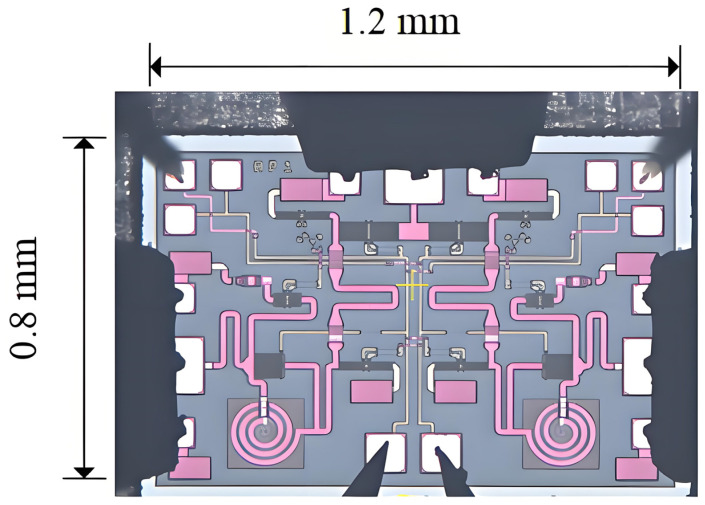
Chip-on-chip test zig-zag diagram.

**Figure 19 sensors-25-03209-f019:**
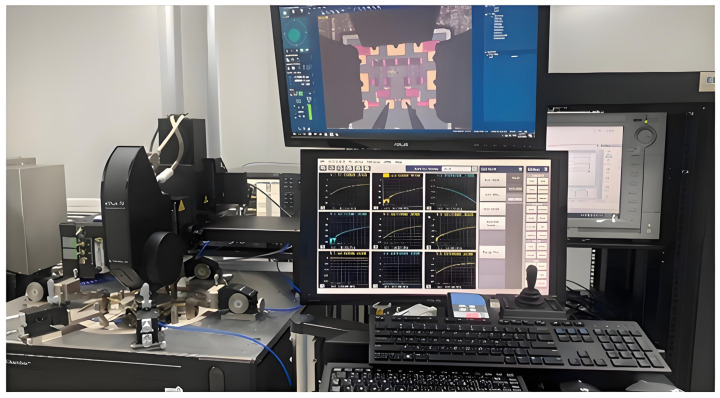
On-chip test environment.

**Figure 20 sensors-25-03209-f020:**
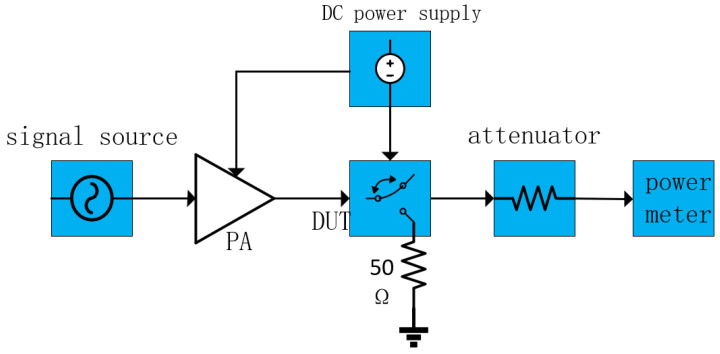
Power Test Topology.

**Figure 21 sensors-25-03209-f021:**
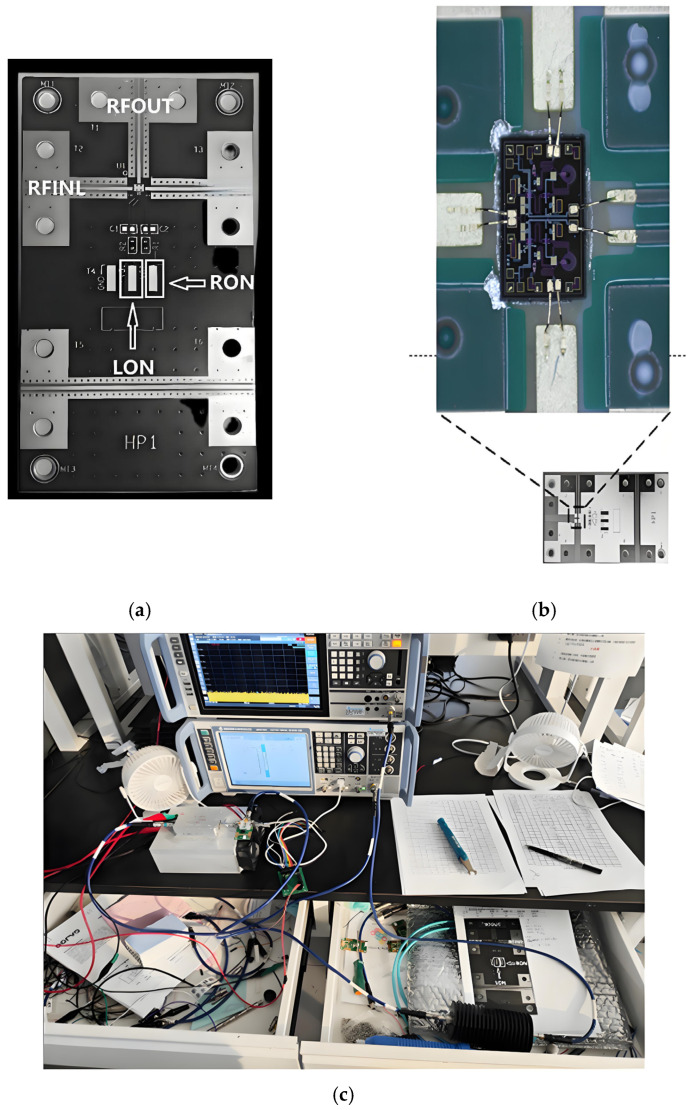
(**a**) Chip assembly board of this paper (**b**) Bonding photo (**c**) Assembly test environment.

**Figure 22 sensors-25-03209-f022:**
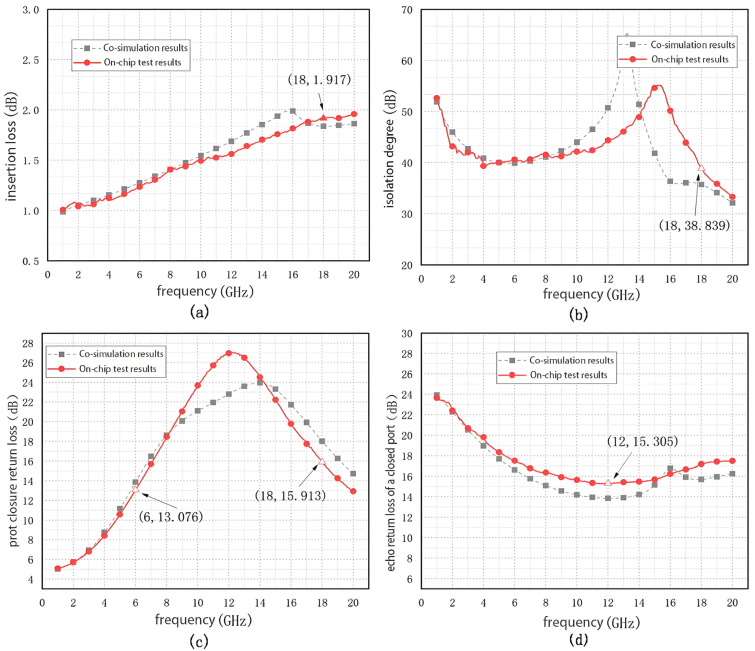
On-chip test results: (**a**) Insertion loss, (**b**) Isolation, (**c**) Switch-off port return loss, (**d**) Switch-on port return loss.

**Figure 23 sensors-25-03209-f023:**
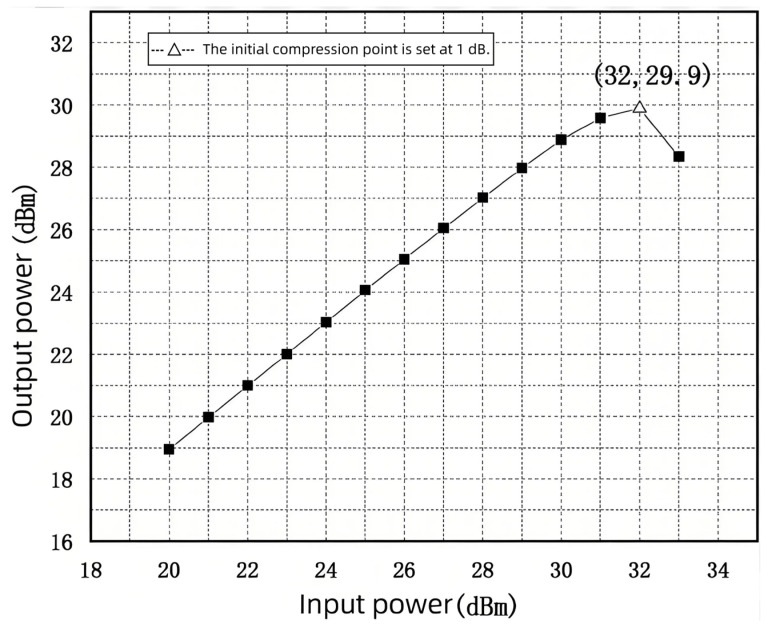
Assembly test simulation results.

**Table 1 sensors-25-03209-t001:** Main device dimensions of chip.

Device Name	Sizes	Device Name	Parameter Size
M1, M2, M10, M11	6 × 70 µm	L1, L2	1.8 nH
M3, M4, M12, M13	2 × 55 µm	C1, C2	2.4 pF
M5, M6, M14, M15	2 × 60 µm	R1, R2	26.2 Ω
M9, M18	8 × 70 µm	Gate Resistor	4.4 kΩ

## Data Availability

No new data were created or analyzed in this study. Data sharing does not apply to this article.

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
