# Peer review of "A Wideband and High-Power RF Switching Design"

_sensors, 2025, doi:10.3390/s25103209_

Round 1

Reviewer 1 Report

Comments and Suggestions for Authors

This paper presents an RF switch chip operating in the 6-18 GHz frequency range, demonstrating a return loss better than 13.075 dB, insertion loss below 1.917 dB, isolation exceeding 38.839 dB, and a power handling capability of 32 dBm. While the work shows promise, several critical issues need to be addressed before it can be considered for publication.

Several figures in the manuscript suffer from low resolution and lack visual clarity. In particular, Figure 1 employs a visual style that is inconsistent with subsequent schematic diagrams (e.g., Figure 2), which affects the overall coherence and professionalism of the presentation.

Table 1 contains several discrepancies. References [41], [44], and [47] are cited within the table but do not appear in the reference list. Additionally, the row labeled "topological structure" incorrectly includes the terms "Return loss (dB)" and "data", which seem miscategorized. Furthermore, the stated operating bandwidth of “DC-20 GHz” conflicts directly with the 6-18 GHz range emphasized throughout the rest of the paper. These inconsistencies should be resolved to ensure clarity and accuracy.

The analysis provided in lines 127-145 focuses solely on insertion loss and isolation for parallel configurations involving 1-3 transistors. For completeness, a similar analysis of series configurations (also involving 1-3 transistors) should be included. This should be supported by corresponding simulation or experimental data, along with comparative discussion.

The simulation process described in lines 295-302 includes excessive and unnecessary detail. This section should be streamlined or partially removed to enhance readability and maintain technical focus without compromising rigor.

The explanation provided in lines 229-236 regarding resonant frequency selection is insufficient. The statement that “resonant points are reasonably controlled within the mid-band of 6-18 GHz” lacks supporting methodological detail. The authors should explicitly specify the criteria used for selecting resonant frequencies and, if possible, include a quantitative comparison of performance across alternative resonant frequency configurations.

Figure 12 is visually cluttered due to overlapping control lines, which hampers readability. The signal paths should be distinctly highlighted, using techniques such as color coding or varied line weights, to improve interpretability and visual clarity.

Reviewer 2 Report

Comments and Suggestions for Authors

1. Some of the pictures (as shown in Figure 21) are not clear enough. It is recommended to insert the original pictures.

2. Is the device in Figure 18 the same as that in Figure 17.

3. The references in Table 1 are not provided at the end. It is suggested to supplement them. It is suggested to compare the relevant research literature in recent years. Some references are repetitive. It is recommended to review them carefully.

4. Whether the row "topological structure" in Table 1 is written incorrectly.

5. The structural dimension diagram of the switch is not specifically provided in the text. It is suggested to supplement it

6. Some chart numbers are confused (for example, "Figure 8 (a)" is mentioned in line 243, but Figure 8 is actually a functional series pipe structure). It is recommended to check the chart numbers throughout the text to ensure accurate correspondence between text and images and avoid confusion among readers.

Reviewer 3 Report

Comments and Suggestions for Authors

This paper presents the design of a wideband and high-power RF switch operating in the 6–18 GHz frequency range.
The logical approach to the design is relatively well written; however, there are several concerns regarding the measurement results, and the quality of the figures in the paper needs improvement.

1. The overall quality of the figures is very low. All text within the figures should be corrected and made more legible.

2. The component values and transistor's size used in Figure 12 should be clearly indicated.

3. In Figure 21, the isolation results raise questions about proper operation in the 6–18 GHz range. The isolation appears to be around 10 dB at 6 GHz, which may not be sufficient.

4. Table 1 should include a comparison of the Figure of Merit (FoM), including the physical size of the design.

5. Is there any performance variation between ports? If so, it would be beneficial to include this in the measurement results.

Reviewer 4 Report

Comments and Suggestions for Authors

Please consider make abstract more concise on key contribution of this work. The novelty aspect of this work should be emphasized, so please consider expand the discussion how this work is differentiated from similar works and consider list each contributions. 

Please include actual data or waveform from assembly testing, and please include better resolution image for key schematic and figure (9, 11, 21, etc). 

Please explain the differences between co-simulation and on-chip test results and provide a more comprehensive discussion about the key parameters contribute to these differences. 

Please add discussion of limitation and provide key insight about practical implementation/application and potential future work. 

Round 2

Reviewer 1 Report

Comments and Suggestions for Authors

The authors have made significant efforts to address the reviewers' comments, particularly in clarifying the methodology and implementing formatting revisions. These improvements have significantly strengthened the technical rigor and presentation quality of the work.

1、On page 14 (Table 2), while the revision from ‘topological structure’ to ‘Circuit topology’ is noted, but the authors should elaborate in detail on the correlation between "Return loss (dB)" and "data" with the ‘Circuit topology’.

2、On page 4 (lines 142-149),  while the experimental constraints (laboratory conditions and tight revision deadlines) are fully acknowledged, it is recommended to supplement the manuscript with simulation analyses of series configurations (involving 1-3 transistors) .
